# Proteins from Blackberry Seeds: Extraction, Osborne Isolate, Characteristics, Functional Properties, and Bioactivities

**DOI:** 10.3390/ijms242015371

**Published:** 2023-10-19

**Authors:** Shaoyi Wang, Fengyi Zhao, Wenlong Wu, Lianfei Lyu, Weilin Li

**Affiliations:** 1Co-Innovation Center for the Sustainable Forestry in Southern China, College of Forestry, Nanjing Forestry University, Nanjing 210037, China; aszx1999@126.com; 2Jiangsu Key Laboratory for the Research and Utilization of Plant Resources, Institute of Botany, Jiangsu Province and Chinese Academy of Sciences, Nanjing 210014, China; zhaofengyi92@163.com (F.Z.); 1964wwl@163.com (W.W.); njbglq@163.com (L.L.)

**Keywords:** seed proteins, blackberry, antioxidant, anticancer

## Abstract

Blackberry fruit contains high levels of nutrients and phenolic compounds. Blackberry pomace accounts for 20~30% of its whole fruit during processing and is generally treated as fertilizer. Blackberry pomace has many seeds that contain carbohydrates, polyphenols, flavonoids, pectin, protein, and other bioactive nutrients. However, its functional properties and seed protein compositions have not been reported. We used a single-factor experiment, response surface, and Osborne isolate method to extract protein isolate, albumin, globulin, glutelin, and prolamin from blackberry seeds for the first time and evaluated their characteristics and functional properties. Glutelin and protein isolate showed good water-holding capacity, emulsification, and foaming capacity, while albumin and globulin showed good oil-holding capacity and thermal stability. They were found to have good antioxidant activities that might be good DPPH free radical scavengers, especially prolamin, which has the lowest IC_50_ value (15.76 μg/mL). Moreover, globulin had the lowest IC_50_ value of 5.03 μg/mL against Hela cells, 31.82 μg/mL against HepG2 cells, and 77.81 μg/mL against MCF-7 cells and a high selectivity index (SI), which suggested globulin had better anti-cervical, antihepatoma, and anti-breast activity but relatively low cytotoxicity. These seed proteins may have great prospects for the development and application of food and drugs in the future.

## 1. Introduction

Protein is the basis of life activity and the organic matter of cells, and it plays a regulatory role in the metabolism of organisms [1]. Protein is divided into plant protein and animal protein according to the food source [2]. Meat, eggs, and milk provided by animals are important sources of animal protein, and the intake of plant protein is mainly through the protein in seeds, such as beans, grains, nuts, and so on [3]. In recent years, plant protein has attracted more attention because of its wide range of sources, short production cycle, and low price. Plant protein is much lower in saturated fatty acids and calories than animal protein, and, more strikingly, it does not contain cholesterol. The intake of more plant proteins than animal proteins can reduce the risk of cardiovascular disease, diabetes, and cancer and can even benefit weight management [4,5]. Studies have also shown that plant protein plays an important role in human health and nutrition, which contributes to its demand as a functional component in food processing [6].

Seed protein is a kind of plant protein with high quality and has been reported as having antibacterial [7,8], immune [9], antioxidant [10,11], anti-hypertension [8,12], anti-hyperlipidemia [13], antiproliferation [14], and other bioactivities [5,15]. Seed protein has many important functional properties, such as emulsifying stability, foaming stability, gelling property, water- and oil-holding capacities, etc. These characteristics can effectively enhance the quality and preservation time of food and even nutritional characteristics that improve food taste, increase elasticity, retain water, and absorb oil [16]. 

Blackberry (*Rubus* spp.) is a kind of small berry fruit in the rose family. It has drawn extensive interest in good biological activities [17], including antioxidant [18], anti-inflammatory [19], antibacterial [20], anticancer [21], and others because of the high content of anthocyanins. This berry is known as “life fruit” and is usually consumed in fresh or processed forms [22]. The process of processing juice, fruit wine, fruit vinegar, and other conventional products will produce a large number of pomaces that account for 20~30%, which is generally discarded or treated as fertilizer, not only causing environmental pollution but also wasting natural resources [23]. Previous studies have found that blackberry pomace has many seeds, which contain carbohydrates, polyphenols, flavonoids, pectin, protein, and other bioactive nutrients, and some components are even higher than the content of pomace and juice [24]. However, the functional properties and seed protein composition of blackberry pomace have not been reported.

In practical applications, seed proteins with high purity are usually extracted from the defatted plant meal. The main extraction techniques of seed protein include conventional (aqua-based, solvent, salt, alkali, detergent) and non-conventional (ultrasonic, high pressure, enzymes, microwave, homogenization, pulse-field) methods [25,26]. Conventional methods may cause lower extraction yields due to protein degradation that is related to a number of factors, including solvents, pH, extraction time, and temperature [27]. Optimizing the protein extraction process can improve protein yield and nutritional and functional properties. Important physicochemical and functional properties of proteins are influenced by a variety of internal and external factors, such as the composition of amino acids, protein conformation, size, and surface properties, including solubility, foaming capacity, and oil absorption [28]. In this work, the classic alkali extraction, acid precipitation, and improved Osborne methods were used to extract the protein from blackberry seeds (Figure 1). Under this method, the protein extraction efficiency was mainly affected by the solid–liquid ratio, temperature, pH, and time, which were assisted with a single-factor experiment and response surface. We further exploited the functional properties of these proteins to seek the nutritional value and advantages of blackberry seed protein in food processing. Finally, we investigated its antioxidant and anticancer activity to provide a theoretical basis for the development of blackberry seed protein in the field of health care. As a by-product of blackberry processing, we hope to improve the effective utilization of blackberry seeds, increase the diversity of seed protein resources, and explore the value of blackberry seed protein in food and health fields.

## 2. Results and Discussion

### 2.1. Single-Factor Test for Seed Protein Extraction from Blackberry·

Four factors were used to analyze the extraction process, including solid–liquid ratio, pH, extraction time and extraction temperature. The results were shown in Figure 1.

In the range of 1:5 to 1:20 g/mL, the solid–liquid ratio was positively correlated with the dissolution of blackberry seed protein. When the solid-to-liquid ratio was increased from 1:5 g/mL to 1:10 g/mL, the extraction efficiency was nearly doubled. From 1:10 g/mL to 1:20 g/mL, the trend of increased extraction efficiency decreased slightly, and the maximum extraction efficiency was 9.13% when the solid–liquid ratio reached 1:20 g/mL. When the solid–liquid ratio reached 1:25 g/mL, the extraction efficiency of blackberry seed protein decreased. Thus, the optimal solid–liquid ratio for extraction of blackberry seed protein by alkaline solution and acid precipitation was 1:20 g/mL.

The extraction efficiency of blackberry seed protein gradually improved with the increase in pH value, and the change was obvious at pH 7~8 and pH 9~11. That may be because the interaction force between sodium hydroxide solution and blackberry seed protein is relatively large at this pH value, and a high pH value will cause high electrostatic repulsion between amino acid components and higher solubility of proteins in alkaline media [29]. When the pH value was 8~9, the extraction efficiency had a relatively slight change, which may indicate that the interaction between proteins is relatively strong at this pH value. This is due to the increasing interaction between sodium hydroxide solution and blackberry seed protein under the condition of increasing alkalinity. In general, the interaction between the proteins gradually weakened, and they were more soluble in sodium hydroxide solutions. Although the extraction efficiency of protein was still increasing, when the pH value exceeded 10, the protein solution was obviously brown-black, and the protein had a certain denaturation under the action of strong alkali. In order to prevent protein denaturation resulting in inactivation, the optimal extraction pH value of this experiment was selected as 10.

From 30 min to 90 min, the protein extraction efficiency was enhanced from 8.45% to 12.36% with the increase in time and reached the maximum value in 90 min. However, after the time exceeded 90 min, the change in protein extraction efficiency tended to be flat with a certain decline. The protein dissolved in the sodium hydroxide liquid over time. Subsequently, the solubility of protein in the sodium hydroxide solution tended to be in a saturated state, so the extraction efficiency did not increase. If the extraction time continues to improve, it will waste resources, reduce the efficiency of extraction, and may reduce the protein extraction efficiency. The optimal extraction time was 90 min.

When the extraction temperature was below 50 °C, the extraction efficiency of protein increased from 7.35% to 9.01% with the increase in temperature; thus, the temperature was positively correlated with the extraction efficiency of protein. This may be because of the increase in temperature, the three-dimensional structure of blackberry seed protein changes, and the molecule of blackberry seed protein becoming more extended, which makes the protein come in full contact with sodium hydroxide solution and results in the improvement of the extraction efficiency of blackberry seed protein. At the same time, the increase in temperature is conducive to the diffusion of protein molecules and more conducive to the dissolution of blackberry seed protein [30]. However, when the temperature exceeded 50 °C, the protein extraction rate decreased, which was due to the denaturation of enzymes and the aggregation of soluble proteins.

### 2.2. Response Surface Methodology to Optimize the Extraction

The results of blackberry seed protein extraction were evaluated according to the experimental conditions shown in Table 1 and Table 2. 

Based on the Box–Behnken design, experiments were conducted to investigate the effects of temperature, time, and the solid–liquid ratio on the content and yield of blackberry seed protein isolate. As shown in experimental results of Table 2, regression analysis was performed on the experimental data using Design-Expert 10.0 software, and the response values and mathematical regression models for each factor were fitted.

It can be seen that the fit lock is not significant (*p* = 0.3302 > 0.05), indicating that the regression equation did not lose fitness; the selected model is suitable and could be used for the fitting test. The equation model was very significant (*p* = 0.0001 ** < 0.01), suggesting that the multiple regression equation could fit the experimental results better. In this model, the significance test of the regression equation revealed that the first-order terms B and C reach the significance level. The main term A and the square terms A^2^, B^2^, and C^2^ reached a very significant level. The order of influence of three factors on the extraction efficiency of blackberry seed protein was A > B > C. The quadratic terms are fitted by software.

The second-order polynomial equation for the composite score (Y) was as follows: Y = 158.07 + 11.75A + 3.24B − 2.06C − 0.85AB + 0.3AC − 0.88BC − 12.87A^2^ − 13.25B^2^ − 8.45C^2^. According to the regression equation, a three-dimensional response surface was constructed to interact with various factors between temperature (A), time (B), and the solid–liquid ratio (C). As shown in Table 2, the F-value of this model was 115.98 (*p* < 0.0001 **), and the significance of this model was extremely high. After fitting, the figure of response surface curves showed similar patterns, and each response surface curve had a maximum point in the experimental region, indicating that the selected range of factors was reasonable. At the same time, higher temperatures, longer extraction times, or higher solid–liquid ratios promoted the protein content and yield of blackberry seed protein. In the response surface, the optimal values for the three variables that maximize both responses simultaneously were a temperature of 51.11 °C, a time of 78.33 min, and a solid–liquid ratio of 1:19.33. The extraction content of blackberry seed protein was 160.40 mg. Taking into account the simplicity of the actual condition, the process parameters were adjusted so that the temperature was 50 °C, the time was 78 min, and the material-to-solvent ratio was 1:19. Under these conditions, the experimental results showed that the extraction content of blackberry seed protein was 159.5 mg and the relative error was 0.31 mg, which suggested the established response surface method model was accurate and effective for the optimization of blackberry seed protein extraction process.

Protein extraction can be traced back to Rustom [30] using the response surface method to investigate the protein extraction from peanut (*Arachis hypogaea* L.). They found time, temperature, pH, and solid–liquid ratio had significant effects on protein extraction and concluded that the optimal extraction conditions were a pH of 8, time of 50 min, temperature of 50 °C, and solid–liquid ratio of 1:8. Similar conclusions were also drawn for the extraction of seed proteins from eggplant [31], flaxseed [32], watermelon seed [33], etc. The optimization of the extraction process of blackberry seed protein will provide a theoretical basis for the future production of blackberry seed protein.

### 2.3. Amino Acid Composition of Blackberry Seed Protein

The glutamate content of blackberry seed protein is 242.54 g/kg, which is the dominant amino acid. According to the research [34], proteins containing high levels of glutamate have the ability to stimulate human immunity and the central nervous system. In addition, glutamate is considered to have good antioxidant capacity because excess electrons can interact with free radicals. Blackberry seed protein showed a high content of acidic amino acids (332.13 g/kg), and the content of hydrophobic amino acids was also at a high level, which made the protein structure more compact, reduced the exposure of hydrophobic regions, and avoided protein aggregation [35]. Hydrophobic amino acids improved lipid solubility and ease of passage through cell membranes, thus showing excellent antioxidant activity [36]. In order to investigate the quality of its nutrition, the essential amino acids of blackberry seed protein were compared with those of the World Health Organization/Food and Agriculture Organization. With the exception of the undetermined tryptophan and methionine (12.45 g/kg), the levels were higher than the World Health Organization recommendation for adults. Blackberry seed proteins are rich in essential amino acids (Table 3), which can be used as an important source of essential amino acids for the food industry and adult dietary fiber.

### 2.4. Sodium Dodecyl Sulfate–Polyacrylamide Gel Electrophoresis (SDS-PAGE)

As shown in Figure 2, the molecular weights of albumin, globulin, glutenin, and protein isolate were detected. It was obvious that there was a large difference in their molecular weight. The protein isolate had major bands at 45 KDa, 40 KDa, 36 kDa, 35 kDa, and 16 kDA, which were similar to Durda’s blackberry seed protein profile [37], with obvious bands around 40 kDa and 45 kDa. It could also be verified that these two subunits can be used as characteristic peptides of the total protein of blackberry seed. The molecular weights obtained were also less than 50 kDa, which was similar to the molecular weights of Persian orange seeds [38] and raspberries [37]. Due to the different extraction conditions, more abundant bands appeared under this condition, which proved that the molecular weights of protein isolates extracted under different conditions were different, but the characteristic bands did not change. There were also two secondary bands of total protein between 55 and 75 kDa. There were major bands of albumin in 170 kDa, 120 kDa, 100 kDa, 90 kDa, 50 kDa, and 38 kDa and more minor bands between 15 and 37 kDa. The prolamin did not show bands between 15 kDa and 170 kDa, possibly because its molecular weight was not in the range or it did not show its molecular weight under reduction conditions. The molecular weight of rice prolamin was only 10–15 kDa [39]. Different molecular weight maps of Gao were obtained under non-reducing and reducing conditions [40]. Yang also showed this phenomenon in the extraction of hempseed prolamin, which did not show bands at 5–270 kDa [41]. Glutelin showed obvious bands only at 36 kDa and 15 kDa. It could be seen that under this condition, the molecular weight distribution of glutelin was relatively simple, which was similar to that of rapeseed glutelin [42].

### 2.5. Functional Activities

#### 2.5.1. Water- and Oil-Holding Capacity

The water-holding capacity of protein from blackberry seeds, albumin, prolamin, glutelin, and BPI were 2.01 ± 0.31 g/g, 0.92 ± 0.01 g/g, 1.56 ± 0.09 g/g, 3.01 ± 0.001 g/g, and 3.61 ± 0.03 g/g, respectively. As shown in Figure 3a, the protein isolate (BPI) showed the best water-holding capacity compared with soy protein isolate with 5.92 ± 0.23 g/g. Other reports included sweet pepper seed protein isolate (2.78 ± 0.02g/g) [43], red bean protein isolate (3.00 g/g) and black bean protein isolate (2.90 g/g) [44], cowpea protein isolate (2.20 g/g) [45], and peanut protein (1.0–2.5 g/g) [46], etc. Previous studies proposed that the water-holding capacity of peanut protein applied in sticky food was 1.49–4.72 g/g. All proteins can be used in sticky foods except globulin, and the higher water-holding capacity of glutelin and protein isolates may be due to the presence of more hydrophilic amino acids, such as glutamate. The glutamate content of the isolated protein was 24.25 g/100 g, which was relatively high [47]. Therefore, glutelin and protein isolate can be used in foods with high water-holding capacity. The water-holding capacity of globulin is less than 1, indicating that the globulin has a lower content of hydrophobic amino acids and a higher water solubility.

The oil-holding capacity (Figure 3b) indicated the ability of the non-polar side chain to bind fat of the protein, and it affected the shelf life and flavor-holding capacity of food [48]. OHC may be affected by the type, charge, hydrophobicity, and oil used of protein [49]. Compared with soy protein isolate (2.94 ± 0.01 g/g), the oil-holding capacity of protein from blackberry seeds, albumin, prolamin, glutelin, and BPI was higher than that of soy protein isolate. Among them, the oil-holding capacity of albumin reached the highest (9.34 ± 0.2 g/g), and its interaction with lipids was strong. The result suggested all these proteins from blackberry seeds have good fat-binding properties and can be used in meat, cakes, and other foods.

#### 2.5.2. Emulsification and Emulsifying Stability

Protein emulsification (EAI) generally reflects its ability to prevent flocculation and aggregation of emulsion droplets by rapidly adsorbing to the oil or water interface during emulsion formation. The emulsion stability index (ESI) is defined as the ability to form a stable emulsion without agglomeration and flocculation forming over time [1]. 

From Figure 4a, we found that gluten (77.14 m^2^/g) and protein isolate (66.67 m^2^/g) have higher emulsification under the condition of alkaline. Moreover, protein isolates (41.45 min) showed the highest emulsification stability; it may be due to a more even ratio of hydrophilic and hydrophobic protein species. Therefore, blackberry seed protein isolate can be used as an alternative emulsifier in the food industry.

#### 2.5.3. Foaming Capacity and Foaming Stability

The foaming capacity of protein is affected by many factors, including protein composition, processing method, temperature, pH, solubility, protein concentration, mix time, and foaming method. At any given pH, better foamability is due to the high flexibility of the protein, which is able to diffuse more quickly to the air–water interface to encapsulate the air, thus enhancing foamability [50]. It plays a key role in various food production, such as whipped toppings, souffle, mousse, meringue, and angel cake [51]. Under the condition of pH 9, the protein isolate showed the best foamability (70% ± 4.5%), followed by the prolamin glutelin. The foaming capacity of albumin and globulin is similar and low (Figure 4b). This may be related to reduced solubility and electrostatic charge of the protein, higher hydrophobicity, and reduced flexibility of the protein.

#### 2.5.4. Thermal Stability

DSC can provide structural stability information based on the endothermic and exothermic processes of the protein. The DSC of albumin, globulin, prolamin, glutelin, and protein isolate (BPI) from blackberry is shown in Figure 5. The larger endothermic peak shows protein denaturation. 

It is well known that the denaturation peak of T_d_ reflects the temperature at which protein denaturation occurs, and ΔH reflects the range of ordered structures [52]. Thermal properties are important to those hydrogen bonds that maintain the integrity of the tertiary structure [53]. Thermal properties can provide a better understanding of the tertiary conformation of proteins so as to be applied to the thermal processing of proteins in the food industry [54].

These five proteins all showed an absorption peak, as shown in Figure 6, which suggested that albumin, globulin, prolamin, and glutelin had different thermal denaturation temperatures, and blackberry protein isolate showed the highest thermal denaturation temperature (globulin (97.42 °C) > albumin (95.63 °C) > glutelin (95.33 °C) > protein isolate (92.62 °C) > prolamin (81.35 °C)). The lowest denaturation temperature of prolamin might be due to the weak hydrogen bond interaction that maintains the tertiary structure or the high flexibility of the tertiary structure of prolamin [53]. In enthalpy, prolamin was 29.94 J/g, similar to that of banana peel prolamin [55], while albumin (55.33 J/g) and globulin (59.2 J/g), glutelin (75.52 J/g), and protein isolate (70.21 J/g) had similar enthalpy. Whether it is the protein isolate or composition, its thermal stability is relatively high, which is caused by different measurement methods. Solid state determination will have relatively high thermal stability, which can also fully understand the stability of protein during dry storage [54,56]. ΔH indicates the degree of ordered structure; however, prolamin has the highest ordered secondary structure and the lowest enthalpy. This may be related to the molecular weight, grade structure, and species of prolamin. Studies have shown that different extraction conditions make the ionic strength and structure of proteins greatly different.

### 2.6. Biological Activities

#### 2.6.1. Antioxidant Activity Assay In Vitro

The ability of antioxidants was evaluated by the determination of DPPH scavenging capacity, which was used to detect a substance’s ability to provide electrons or hydrogen, thereby converting free radicals into more stable substances [57]. As shown in Figure 7, with the increase in the concentration of protein solution, the ability of DPPH to remove free radicals also increased. When the sample concentration was 0.05 mg/mL, the clearance rate of prolamin tended to be stable, and when the sample concentration was 0.1 mg/mL, the clearance rate of protein isolate tended to be stable. When the concentration was 0.05 mg/mL, the clearance rate was as follows: prolamin > protein isolate > albumin > globulin > glutelin. However, at 0.2 mg/mL, the clearance rate of glutelin was greater than that of albumin and globulin. As seen in Maha’s report, a similar situation was found in the clearance of glutelin [58]. This might be related to protein structure, composition, and hydrophobicity [59]. As calculated in Figure 7a, the half inhibitory concentration (IC_50_) of albumin, globulin, prolamin, glutelin, and protein isolate was 30.37, 72.66, 15.76, 46.40, and 78.28 μg/mL. It suggested that prolamin had the lowest IC_50_ value while protein isolate had the highest IC_50_ value. These results indicated that there might be a large number of hydrogen donors in the prolamin and protein isolate of blackberry seed protein, which could react with free radicals to produce more stable products and terminate the free radical chain reaction. The results showed that blackberry seed protein, especially prolamin, could be a good DPPH free radical scavenger.

The cationic free radical scavenging ability of ABTS is a measure of the total antioxidant capacity of a substance. Under the action of oxidizing agents, ABTS are oxidized to stable blue-green cationic radicals. As shown in Figure 7b and in the inhibition of superoxide anion in Figure 7c, prolamin showed the best ABTS cationic radical scavenging ability, followed by protein isolate and glutelin. Compared with DPPH clearance, glutelin showed better ABTS clearance capacity. Both prolamin and protein isolate had relatively good free radical scavenging ability and may act as potential antioxidants. In terms of anti-superoxide anion ability, the lowest anti-superoxide anion ability was globulin with 52.12 ± 1.21 U/g, while the highest anti-superoxide anion ability was prolamin with 72.01 ± 0.82 U/g, and the other proteins had similar abilities. Three kinds of antioxidant determination showed that the protein isolated and the other four grading proteins had a certain resistance to oxidation, and prolamin had the best oxidation resistance performance.

#### 2.6.2. Anticancer Activities and Cytotoxicity Assay In Vitro

Anticancer activities of albumin, globulin, prolamin, glutelin, and protein isolate (BPI) from blackberry were evaluated on the growth of the human cancer cell lines Hela, HepG2, MCF-7, and A549 and normal cells (HUVECs) by MTT assay at the concentration of 6.25, 12.5, 25, 50, 100, 200, 400 μg/mL in vitro. The results are shown in Table 4. 

It is well known that the lower the IC_50_ value, the higher its activity. Globulin had the lowest IC_50_ value of 5.03 μg/mL against Hela cells, 31.82 μg/mL against HepG2 cells, and 77.81 μg/mL against MCF-7 cells, which suggested globulin had the highest anti-cervical, antihepatoma and anti-breast activity against Hela, HepG2 and MCF-7 cells. Glutelin (79.84 μg/mL) had higher anti-cervical activity than albumin (107.42 μg/mL), protein isolate (153.28 μg/mL), and prolamin (385.75 μg/mL). Cervical cancer is a common cancer and the fourth leading cause of cancer death in women. Long-term infection with human papillomavirus (HPV) is one of the causes of cervical cancer, and approximately 90.83% of cervical cancer patients are infected with high-risk HPVs [60,61]. Similar with cervical cancer, breast cancer is one of the most common malignant tumors among women. Protein isolate had higher antihepatoma (108.82 μg/mL) and anti-breast (109.89 μg/mL) activity than albumin, glutelin and prolamin. Liver cancer is a global health challenge, with an estimated incidence of more than 1 million cases by 2025. Hepatocellular carcinoma (HCC) is the most common form of liver cancer, accounting for about 90% of cases. Hepatitis B virus and hepatitis C virus infection are major risk factors for HCC [61]. Glutelin (79.84 μg/mL) had the highest anti-lung activity than the other four proteins. Lung cancer is the most frequently diagnosed of all cancer types and the leading cause of cancer-related death, with an estimated 8.16 million confirmed cases of lung cancer and 0.715 million deaths from lung cancer in China [62]. HUVEC cells are nonmalignant cell lines that are used to evaluate the cytotoxicity and characterize the selectivity described as selectivity index (SI) (SI = (IC_50_ for nonmalignant cell line HUVEC)/(IC_50_ for human tumor cell line)), displayed in Figure 8. An SI value of more than 1 indicates low toxicity, and the higher the SI value, the lower the toxicity.

The evaluation of SI is an important means for determining the toxicity of drugs in future pharmacological applications. All these five blackberry seed proteins had moderate to good cytotoxic activity against human cancer cells, and almost all proteins had an SI value of more than 1, especially protein isolate and globulin. As reported, protein drugs own unique advantages in cancer treatment compared to chemotherapy and gene therapy. Protein drugs have high efficiency and specific anticancer effects directly, which can induce apoptosis of cancer cells through clear signaling pathways or indirectly inhibit tumors by regulating tumor microenvironment or stimulating immune response [63]. 

Protein isolate, albumin, globulin, prolamin, and glutelin from blackberry seeds had higher anticancer activities against different cancer cells but relatively low cytotoxicity, which suggested blackberry seed proteins may have great prospects for the development and application of food and drugs in the future. 

## 3. Materials and Methods

### 3.1. Materials

Seeds of blackberry were taken from Institute of Botany, Jiangsu Province, and the Chinese Academy of Sciences, Baima field. These seeds were preserved in refrigerator at −30 °C. MTT, DMEM, FBS, and penicillin/streptomycin were all commercially purchased from Nanjing Keybionet Biotechnology Co., LTD., Nanjing, China. Hela, HepG2, MCF-7, A549, and HUVEC cells were purchased from the National Collection of Authenticated Cell Cultures. All solvents and chemicals used in the experiments were of analytical or chromatographic grade unless otherwise stated. 

### 3.2. Extraction of Protein of Blackberry

#### 3.2.1. Degreasing and Preparation of Protein Isolates

First, sieve the blackberry seed powder through 60 mesh. To remove fat, blackberry seed powder was mixed in a 1:1 ratio of petroleum ether, filtered (repeated three times), and air-dried in a ventilated dry place. The protein isolate was extracted using the method of alkali extraction and acid precipitation from 100 g of blackberry seed powder each time under the optimal parameters of response surface optimization.

The protein components were extracted according to Osborne method classification described by Adebiyi with slight modifications. First, 100 g of blackberry seed powder was mixed with 1 L of distilled water at room temperature under magnetic stirring for 2 h. Then, they were centrifuged (6000 rpm, 10 min) to obtain albumin extract. The residue was extracted with NaCl (0.5 M, 1 L) and 1 L of 75% ethanol, and NaOH (0.1 M, 1 L) was used to adjust to pH 10. After mixing under magnetic agitation for 2 h, it was centrifuged, and globulin, prolamin, and glutelin extract solutions were obtained, respectively. The supernatant was concentrated by vacuum rotary evaporation to remove ethanol. The four protein extract solutions were dialyzed at 4 °C for 72 h. Finally, the protein powder was obtained by freeze drying and kept at −20 °C.

##### The Single-Factor Test

The effects of the ratio of blackberry seed powder to water, time, temperature, and pH value on the protein extraction rate during protein extraction were analyzed by single-factor test. The solid–liquid ratio was selected as 1:5, 1:10, 1:15, 1:20, and 1:25 g/mL, and the time was selected as 30, 60, 90, 120 and 150 min. Temperature was 30, 35, 40, 45, 50, and 55 °C; pH was adjusted by 0.5 mol/L NaOH; pH was selected at 7, 8, 9, 10, and 11; and the best parameters were determined. The results were calculated by the following formula: E (%) = (V·c/m) × 100. E (%) was the extraction efficiency of protein, V (mL) was the volume of the extract solutions, c (g/mL) was the content of protein in the extract solutions, and m (g) was the weight of the protein sample.

##### Response Surface Test Design

In this experiment, Box–Behnken design (BBD) design principle in Design-Expert 10.0 software was used to design response surface tests. On the basis of the single-factor test, three factors, solid–liquid ratio, time, and temperature, were selected as the test factors, and the protein extraction rate was used as the response value. The independent variables included extraction time, temperature, and liquid-to-solid ratio, which were represented by A, B, and C, respectively. Additionally, −1, 0, and 1 represent the three levels of the independent variables. The design test factors and levels are shown in Table 5.

### 3.3. Determination of Concentration of Protein

The protein was determined by Coomassie brilliant blue method. A total of 1 mL of supernatant was taken and diluted to a certain multiple, and 0.05 mL of diluent and 3 mL of Coomasi bright blue solution were mixed in test tube stew for 10 min. The absorbance values were determined through enzyme marker (Thermo scientific Spectrophotometer 1530, Shanghai, China) at 595 nm, and the extraction concentration of protein from blackberry seed was further calculated.

### 3.4. Determination of Amino Acid Composition

The blackberry protein isolate (BPI) was accurately weighed and placed in an anaerobic hydrolysis tube and mixed with 5 mL of 6 equivalent hydrochloric acid, then frozen in refrigerant (liquid nitrogen or carbon dioxide ice). After the solutions were solidified, the tube was sealed by vacuum. It was hydrolyzed in a constant temperature drying oven at 110 °C for 13 h. After cooling, the volume was fixed to 10 mL, and a water filtration membrane with a pore size of 0.45 μm was used to remove impurities. A total of 0.5 mL of filtrate was taken into the EP tube and vacuum-dried. The residue was dissolved in 1 mL of deionized water, then dried and repeated twice. Finally, 1 mL of sample diluent with pH 2.2 was added to dissolve, and an aqueous filter membrane with pore size of 0.22 μm was added to filter. The automatic amino acid analyzer (S-433D, Sykam, Germany) was used for analysis. 

### 3.5. Sodium Dodecyl Sulfate–Polyacrylamide Gel Electrophoresis (SDS-PAGE)

The native molecular weight distribution and protein composition of blackberry protein isolate (BPI), prolamin, glutelin, albumin, and globulin were further identified with size exclusion chromatography (SEC) coupled with SDS-PAGE. According to Laemmli [64], with some modification, the protein solution (50 mg/mL) was prepared by a reducing protein buffer named ZS306, and 20 μL of solution was taken onto the gel. Electrophoresis was performed at a constant current of 80 V for the first 30 min and at a constant current of 135 V for approximately 90 min. The gel is dyed Coomassie brilliant blue using ExBlue protein ultrafast dyeing solution (ZD305A) and decolorized in distilled water.

### 3.6. Functional Activities

#### 3.6.1. Water-Holding Capacity (WHC) and Oil-Holding Capacity (OHC)

WHC and OHC were determined to be slightly modified according to the method described by Tang et al. [65]. In brief, 0.01 g of a dried protein sample (M_0_) was dispersed in 1 mL of deionized water and soybean oil in a pre-weighed centrifuge tube (1.5 mL, M_1_). The dispersion was swirled for 30 s and then centrifuged at 5000 rpm for 15 min. After decanting the supernatant, the tube containing the protein residue (M_2_) was weighed. WHC and OHC (g/g) were then calculated by the following formula: WHC/OHC (g/g) = (M_2_ − M_1_ − M_0_)/M_0_.

#### 3.6.2. Emulsification and Emulsifying Stability

Referring to [1], a 1% (*w*/*v*) protein solution was prepared with deionized water, and the pH value was adjusted to 9. Soybean oil and the protein solution were mixed evenly at a volume of 1:3, homogenized by a homogenizer (10,000 rpm, 2 min), and the homogenized emulsion was 80 µL. Absorbance was measured at 500 nm after 100 times dilution with 0.1% SDS. Emulsifying activity and stability were calculated according to Formula (1) and Formula (2), respectively:(1)$$EAI\;(m^{2}/g)=\frac{2\times 2.303\times A\times DF}{C\times \varphi \times 10^{4}}\times 100$$
(2)$$ESI\;(\min)=\frac{A_{0}}{A_{0}-A_{10}}\times t$$
C = protein concentration; DF = dilution factor (250); 1 = path length (1 cm); φ = oil volume fraction (0.25); A_0_ = absorbance at 0 min; A_10_ = absorbance at 10 min; t = 10 min. 

#### 3.6.3. Foaming Capacity and Foaming Stability

Foaming capacity (FC) and foam stability (FS) were measured according to Hojilla-Evangelista [66]. A pH (9) sample was prepared in a 50 mL centrifuge tube (10.0 mL, 10.0 mg/mL). The sample was homogenized for 2 min at 15,000 rpm with high-speed homogenizer (FSH-2A, Jinwei, Changzhou, China) instrument. FC and FS are calculated as follows:FC (%) = (V_0_ − V)/V; FS (%) = V/V_30_

V, V_0_, and V_30_ refer to the initial volume of the sample and the foam volume at 0 min and at 30 min, respectively.

#### 3.6.4. Thermal Stability

Differential scanning calorimetry (DSC) was used to measure the thermal stability (DSC 214, Netzsch, Gertebau, Germany). The protein sample (BPI, prolamin, glutelin, albumin, and globulin, 5.0 mg, respectively) was accurately weighed onto the aluminum plate. The crucible is sealed and heated at a nitrogen atmosphere in a range of 0 to 300 °C and a scan rate of 10 °C/min. The tests were carried out under nitrogen (N_2_) at a flow rate of 20 mL/s.

### 3.7. Biological Activities

#### 3.7.1. Antioxidant Activity Assay In Vitro

1,1-diphenyl-2-picrylhydrazyl radical (DPPH) scavenging capacity was determined using DPPH (A153-1-1) determination kits (Nanjing Jiancheng Institute of Bioengineering, Nanjing, China). In order to determine the scavenging ability of DPPH free radicals, DPPH was prepared into 0.2 mg/mL protein sample solution with dimethyl sulfoxide and diluted in equal proportion. The free radical scavenging rate and the free radical scavenging rate of each concentration sample were calculated as the value of IC_50_.

Total antioxidant capacity was determined using a T-AOC (A015-2-1) assay kit (Nanjing Institute of Jiancheng Bioengineering, Nanjing, China). ABTS oxidized to green ABTS^+^ under oxidation. The protein sample solution was prepared with dimethyl sulfoxide at 5 mg/mL, and the total antioxidant capacity was determined. In this experiment, the antioxidant capacity of Trolox was expressed as a multiple of the antioxidant capacity of protein. The antioxidant capacity was expressed as Trolox-equivalent antioxidant capacity (TEAC). 

The inhibition of superoxide anion free radical (A052-1-1) assay kit (Nanjing Jiancheng Bioengineering Institute, Nanjing, China) was used to determine the inhibition of superoxide anion. The reaction system of xanthine and xanthine oxidase in the body was simulated to produce superoxide anion free radical, and the electron transport substance and gress chromogenic agent were added to make the reaction system appear as a purply red. The absorbance value was measured at 550 nm by spectrophotometer and compared with V_c_. The samples were prepared with dimethyl sulfoxide as 1 mg/mL protein sample solution, and the superoxide anion capacity was determined.

#### 3.7.2. Anticancer Activities and Cytotoxicity Assay In Vitro

Referred to our previous report [67], pretreatment cells (Hela, HepG2, MCF-7, A549, and HUVEC cells) were incubated with BPI, prolamin, glutelin, albumin, and globulin at different concentrations for 48 h at 5% CO_2_ and 37 °C, with the process being repeated three times for each concentration. The following treatment was consistent with the above-mentioned reports. Finally, cell viability values were calculated using the Origin software (Origin 2019b).

### 3.8. Statistical Analysis

SPSS Statistics 16.0 (IBM, Chicago, IL, USA) software was used for the analysis of data with one-way ANOVA at a 0.05% level of reliability. Origin 2019b (OriginLab Crop., Northampton, MA, USA) was used to analyze the IC_50_ value with linear fitting. All experiments were repeated three times, and the results represented the mean ± standard deviation.

## 4. Conclusions

In summary, we successfully obtained protein isolate, albumin, globulin, glutelin, and prolamin from blackberry seeds for the first time by single-factor experiment, response surface, and Osborne isolate method. The molecular weight of blackberry seed protein isolate had five main bands, and the content of essential amino acids met the recommended standards of WTO except for tryptophan and methionine. The four compositional proteins extracted by the Osborne method showed their own characteristics and advantages. Glutelin and protein isolate showed good water-holding capacity, emulsification, and foaming capacity, while albumin and globulin showed good oil-holding capacity and thermal stability. These five proteins, especially prolamin, were found to be good DPPH free radical scavengers in the evaluation of antioxidant activities. Moreover, higher anticancer activities against different cancer cells but relatively low cytotoxicity suggested these seed proteins could have great prospects for the development and application of food and drugs in the future. We hope it will provide a theoretical basis for the production and application of blackberry seed proteins in the food industry and health care. 

## Data Availability

Data are available upon request.

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
