# Peer review of "Proteins from Blackberry Seeds: Extraction, Osborne Isolate, Characteristics, Functional Properties, and Bioactivities"

_ijms, 2023, doi:10.3390/ijms242015371_

Round 1

Reviewer 1 Report

Wang et al. focused their studies on the extraxtion, by means of different protocols, of protein isolate, albumin, globulin, glutelin and prolamin from blackberry seeds. The paper is within the scope of the journal and the topic is interesting and relevant for the field. Tables and figures are appropriate. The experimental design is appropriate and understandable.

I only have a few minor recommendations:

-Correct, throughout the text, “in vitro” in italics

- When p value is mentioned in the text, include asterisks

- You should add the concentrations used in the MTT assay

- I think that further antioxidant assays should be performed (e.g. ABTS, ORAC etc)

-The antioxidant effect should also be demonstrated with a specific cell-based assay

The article is well written so only a general revision is required.

Reviewer 2 Report

In general the article is well organized and easy to read, the figures are ok, and the discussion could be improved. But, I do not believe that the article makes a relevant contribution to current scientific knowledge.

I detected some mistakes, like:

Line 13: "containing" change to "contain"

Line 71-75: Please rewrite the aim/objective of the article

I think there is no need to include section 2.1. Single-factor test for seed protein extraction from blackberry because there is a more interesting statistical analysis in section 2.2.

Line 154-156: How do the authors find the optimal point? Through desirability function?

Line 304: the caption of Figure 7 doesn’t correspond to the figure.

The English should be checked through all the article

Round 2

Reviewer 2 Report

The authors addressed all the concerns/questions related to the manuscript.